# Impacts of Life-Time Exposure of Arsenic, Cadmium and Fluoranthene on the Earthworms’ *L. rubellus* Global DNA Methylation as Detected by msAFLP

**DOI:** 10.3390/genes13050770

**Published:** 2022-04-26

**Authors:** Ilze Rasnaca, Peter Kille, Lindsay K. Newbold, David J. Spurgeon

**Affiliations:** 1UK Centre for Ecology and Hydrology, Maclean Building, Benson Lane, Crowmarsh Gifford, Wallingford, Oxfordshire OX10 8BB, UK; ilzerasn@gmail.com (I.R.); lise@ceh.ac.uk (L.K.N.); 2Cardiff School of Biosciences, University of Cardiff, P.O. Box 915, Cardiff CF10 3TL, UK; kille@cardiff.ac.uk

**Keywords:** pollution, epigenetic, methylome, phenotypic plasticity

## Abstract

This study reports on the effects of long-term exposure to the metals arsenic (As), cadmium (Cd) and the polycyclic aromatic hydrocarbon fluoranthene on the survival, growth, development and DNA methylation status of the earthworm *Lumbricus rubellus*. Exposures to the three chemicals were conducted over their whole juvenile developmental period from egg to adult. Significant effects on one or more measured endpoints were found for all three chemicals. Arsenic had no effect on survival, but had a significant effect on growth rates at concentrations of 36 mg/kg or higher and also slowed the rate of maturation. Cadmium significantly reduced juvenile survival at 500 mg/kg, juvenile growth at 148 mg/kg and maturation rates at all tested concentrations. Fluoranthene had no effect on survival or the developmental period, but did significantly reduce growth rates at 800 mg/kg. Effects at these concentrations are consistent with the known effects of these three chemicals on earthworms from previous studies conducted mainly with *Eisenia fetida*. Both As and Cd had no effect on DNA methylation patterning in earthworms measured at the end of the exposure. Fluoranthene was shown, for the first time. to have an effect on a species’ DNA methylation levels. These results suggest that apical phenotypic changes for As and Cd are not necessarily associated with changes in DNA methylation profiles. However, exposure to the organic chemical fluoranthene influenced DNA methylation patterns, suggesting wider remodelling of the epigenome for this chemical.

## 1. Introduction

Tests to measure the toxicity of chemicals are generally performed over short time periods lasting from hours to weeks. This does not reflect the nature of real-life toxic exposures, which can last for months to years. These extended exposure times mean that organisms can be exposed to persistent chemicals for a full life time or even for multiple generations. These increased exposure times can lead to greater effects due to the accumulation of the chemical or progression in the accrual of damage [1,2,3]. Alternatively, species may show phenotypic adaptations to the chemical, due to the development of stress hardening traits [4]. It is, therefore, important to explore the long-term impacts of pollution on affected organisms to understand the effects of prolonged exposure. Such studies may also help elucidate how measurements of toxicity made in typical short-term experiments relate to the effects of long-term exposures relevant to field cases.

Beyond their established and characterised effects on population relevant life-cycle traits, long-term chemical exposure may also have impacts on organisms at a molecular level. For instance, exposure to chemicals has been shown to result in changes in chromatin structure, detected via DNA methylation [5]. DNA methylation is commonly the marker of choice for the understanding of epigenetic effects, due to its importance in normal cell functioning [6] and the availability of techniques for studying this mark in exposed organism [7,8]. There are multiple examples of changes in the DNA methylome that have been associated with chemical exposure [9]. However, the nature of methylation change under chronic long-term exposure remains poorly known.

The earthworm species *L. rubellus* has been comparatively widely used in field and laboratory studies on the short- and long-term effects of chemical exposures. Cd has been found to have an impact on *L. rubellus* growth and metallothionein-2 expression [10]. Both Cd and the polycyclic aromatic hydrocarbon fluoranthene have also been found to have an impact on *L. rubellus* reproduction and gene expression [11,12]. Arsenic exposure has been found to both cause detrimental impacts on *L. rubellus* in a short-term exposure and adaptation after prolonged exposure at polluted sites [13]. DNA methylation has been indicated as a possible biomarker when observing multiple metalloid impacts on earthworms in a study by Santoyo et al. [14]. Earthworms are a potential valuable tool for studies of DNA methylation change as, unlike some taxa such as arthropods, they exhibit relatively high levels of DNA methylation comparable with those for some vertebrate species [15]. In one study of *L. rubellus* collected from naturally heavily polluted soils, DNA methylation levels were correlated with As levels in soils in one cryptic lineage of this earthworm morphospecies, while for a second lineage this was not the case [16]. A further study using *Lumbricus terrestris* found that long term exposure to low level Cd resulted in an increase in global DNA methylation, some of which was retained by earthworms placed in clean soil [17]. Taken together, these studies indicate that long-term pollutant exposure can cause changes to the earthworm genome methylation status, with as yet unknown functional consequences.

To establish the responses of the earthworm epigenome and life-cycle traits to chemical exposure, we characterised here the DNA methylation status and effects on growth and development following exposure over the full juvenile period. These studies allowed us to test the hypothesis that changes in the DNA methylation status plays a role in earthworm responses to persistent pollutant exposure. Global DNA methylation, measured by methylation-sensitive amplified fragment length polymorphism (msAFLP), was assessed in *L. rubellus* exposed to three chemicals (the metal Cd, metalloid As and the polycyclic aromatic hydrocarbon (PAH) fluoranthene) in a long-term laboratory test (>140 days). Effects on juvenile survival, growth and maturation were measured over the full developmental period, with DNA methylation also assessed in individuals at the end of the exposure.

## 2. Materials and Methods

### 2.1. Life-Time Exposures of L. rubellus to As, Cd and Fluoranthene

The *L. rubellus’* exposures to As were carried out from hatchling to adulthood over 280 days, using the method of Anderson et al. [18]. Exposures to Cd and fluoranthene were performed in the same manner as for As, except that the Cd test was conducted over 140 days and the fluoranthene test over 308 days. The difference in exposure time was necessary because at the highest concentrations of Cd tested in the experiment some mortality of the earthworms was detected. Therefore, the exposure duration for this metal was shortened to ensure there were sufficient earthworms available for DNA methylation analysis at these higher concentrations. Shortening of the Cd test meant that it was not possible to allow the exposure to continue to full development at the highest exposure concentrations.

For the exposures, a clay loam soil (Broughton Loams, Kettering, UK) amended with 3% organic material (“GroSure”, Westland, Dungannon, UK) was spiked with As (Na_2_HAsO_4_·7H_2_O, Sigma Aldrich, Bournemouth, UK), fluoranthene (99% purity) (Sigma Aldrich) or Cd (applied as CdCl_2_, Sigma Aldrich). Both the As and Cd salts are water soluble and were added to the soil as a stock solution dissolved in sufficient water to give the required 50% water holding capacity needed for the test. Fluoranthene was added as an acetone stock solution to the soil surface. Top-up additional solvent was added to the control and lower concentrations to ensure that all containers (including controls) received a similar amount of acetone. After allowing the residual acetone to evaporate (checked by assessment for residual wetting and smell), the spiked soils were then thoroughly mixed and water added to bring the soil up to 50% of water holding capacity. Unlike As and Cd, fluoranthene is subject to microbial degradation in soil with a measured half-life of 2.4 weeks in coarse peat sand and 36 weeks in compost [19]. Therefore, at 16 weeks, the earthworms exposed to this chemical were placed into newly prepared soil with the appropriate amounts of chemical added to ensure that the concentration did not differ far from nominal values over time.

The concentrations originally used were chosen to cover the full sublethal range of exposures, selected based on Svendsen et al. [11] and Anderson et al. [18]. The concentrations of chemicals (per dry weight soil) used were intended to result in sufficient earthworm survival for later analysis, As: 0, 3, 12 and 36 mg/kg; Cd: 0, 13, 43.9, 148 and 500 mg/kg; fluoranthene: 0, 20.8, 31.2, 70.2 and 800 mg/kg. Individuals exposed to each of these concentrations were used for the msAFLP analysis.

To start the test, 25 young hatchling earthworms were placed individually in 200 mL containers with 250 g of control or spiked soil and 1 g (dry weight) horse manure added to the surface as a source of food. The manure used as food was spiked with the relevant concentration of the test chemical by its addition to the water used for wetting the manure to 80% water holding capacity for As and Cd, or in acetone as described above for the soils for fluoranthene. This ensured both dermal and ingestion exposure. The containers were kept at 14 °C ± 2 °C under constant light for the duration of the experiment. Every 28 days, earthworm survival, weight and developmental status were recorded. At each of these timepoints, after removal of the previous manure, an additional amount of 1 g (dry weight) of appropriately spiked horse manure was added to the soil surface. The amount of manure was increased to 1.5 g once >90% of the earthworms in the control treatment had reached adulthood–indicated by the presence of a fully developed clitellum. At the end of the exposure, the earthworms were recovered from the test soil and final weight recorded. All individuals were then snap frozen in liquid nitrogen and stored at −80 °C. These whole earthworm samples were later ground to powder under liquid nitrogen and used for DNA extraction for the genotyping and msAFLP analysis.

The weight data for the As-, Cd- and fluoranthene-exposed earthworms were tested for normality using the Shapiro–Wilks test and for statistically significant difference between treatments with ANOVA, using the data taken from the final weight measurements made for each chemical (which was obtained at the same time the earthworms were sampled for the msAFLP analysis). Time to maturation was compared using the Kruskal–Wallis test and the Dunn post hoc test. Analyses were carried out in the R statistical environment.

### 2.2. DNA Extraction

DNA was extracted from ground tissue samples from individual earthworms using the Qiagen blood and tissue DNA extraction kit, according to the manufacturers’ instructions (Qiagen Inc., Crawley, UK). The DNA isolated from the extracted tissues was quantified using a NanoDrop spectrophotometer (NanoDrop Technologies, Wilmington, DE, USA) to provide a quantitative measure of DNA concentrations. DNA quality was checked using gel electrophoresis.

### 2.3. Methylation Sensitive Amplified Fragment Length Polymorphism (msAFLP) Analysis

The protocol and downstream assessment used for msAFLP analysis were adapted from Díaz-Freije et al. [20], Ardura et al. [21] and Blouin et al. [22]. Following extraction, the DNA sample was diluted to acquire similar concentrations of DNA in each sample and split into two aliquots. Each was then cleaved using two restriction enzymes, either EcoRI (ThermoFisher Scientific, Birchwood, UK) and HpaII (ThermoFisher Scientific), or EcoRI and MspI (ThermoFisher Scientific).

The DNA samples, appropriate restriction enzymes and CutSmart™ buffer (New England Biolabs, Hitchin, UK) were placed in 37 °C temperature for 3 h to allow the digestion to occur. The linker sequences used for the ligation step (all oligonucleotides were obtained from Sigma Aldrich) were: EcoRI linker 1: CTCGTAGACTGCGTACC; EcoRI linker 2: AATTGGTACGCAGTCTAC; MspI/HpaII linker 1: ACGATGAGTCCTGAG; MspI/HpaII linker 2: TACTCAGGACTCAT. The ligation mix included: the cut DNA produced in the digestion reaction, linkers, T4 DNA Ligase (ThermoFisher Scientific), ATP (ThermoFisher Scientific), CutSmart™ buffer (New England Biolabs, Hitchin, UK) and deionised water. The mix was placed at 37 °C temperature for three hours. After this time, two amplification steps using the Polymerase Chain Reaction (PCR) were performed, the pre-selective PCR and the selective PCR.

The primers used in the pre-selective PCR were: EcoRI-pre: GACTGCGTACCAATTC; Msp-pre: GATGAGTCTAGAACGGA. The primers used in the selective PCR step were: EcoRI-AC: 6-FAM-GACTGCGTACCAATTCAC; EcoRI-AG: 6FAM-GACTGCGTACCAATTCAC; MspI-ATC: GATGAGTCTAGAACGGATC; MspI-AG: GATGAGTCTAGAACGGAG. Both PCR reactions included the following: ligated DNA, deionised water, deoxyribonucleotide mix (Applied Biosystems), Taq DNA polymerase including MgCl2 (Sigma Aldrich), Taq PCR buffer (Sigma Aldrich) and the appropriate primers. The cycling conditions for both reactions were: Pre-selective PCR: 72 °C for 2 min, 20 cycles of 94 °C for 20 s, 56 °C for 30 s, 72 °C for 2 min and a final step of 60 °C for 30 min. Selective PCR: 94 °C for 2 min, 10 cycles of 94 °C for 20 s, 66 °C (decreasing by 1 °C each cycle) for 30 s and 72 °C for 2 min, followed by 20 cycles of 94 °C for 20 s, 56 °C for 30 s and 72 °C for 2 min, ending with 60 °C for 30 min. The resulting PCR products were then cleaned using the ZR DNA Sequencing Clean-up Kit™ (Zymo Research), mixed with GeneScan™ 600 LIZ™ size standard v2.0 (ThermoFisher Scientific) and loaded into an Applied Biosystems 3730 DNA Analyzer to generate the fragment analysis results used for the msAFLP data analysis.

### 2.4. MsAFLP Data Analysis

The AFLP scoring and fragment analysis were performed using GeneMapper v.4.0 software (Applied Biosystems). The resulting scoring was analysed using the R package msap, as detailed in Ardura et al. [21]. Briefly, the numbers of two types of the CCGG restriction sites can be acquired, unmethylated where both restriction enzymes have cut the site, or partially methylated (either the internal or external C is methylated) where one of the restriction enzymes (HpaII or MspI), but not the other, has cut the site. In theory, another type of site, where hypermethylation has occurred and neither enzyme has been able to cut the restriction site, can be detected. However, this may also occur if a polymorphism is present in this site, therefore hypermethylation is likely to be rare, especially given that genetically diverse organisms were used in this experiment.

Four types of sites were identified from the data acquired from msap analysis, these were: Type 1: both HpaII and MspI have cut (two fragments present); Type 2: MspI has cut, HpaII has not; Type 3: HpaII has cut, MspI has not; Type 4: Neither enzyme has cut, either due to polymorphism or hypermethylation. The global methylation level was calculated as the proportion of partially methylated loci over all scorable sites (partially and unmethylated sites, excluding “hypermethylated” sites).

The analysed restriction loci were divided into methylation-susceptible loci (MSL) or non-methylated loci (NML), only those sites showing polymorphisms with at least two occurrences of each state were used. The MSL sites were used for assessing epigenetic polymorphisms and the NML sites for genetic ones. The msap package also carried out Principal coordinates’ analyses (PCoA) and analysis of molecular variance (AMOVA). Shapiro–Wilks test was performed to check for residual normality and the Kruskal–Wallis test was performed to compare groups. Spearman’s rank test was performed to assess correlations. This analysis was completed using Microsoft Office Excel and R statistical software. The msap package was also used to acquire the AFLP scores for genetic variability between the populations by employing the meth(false) option. This option scores all loci for genetic difference, not just NML, therefore providing a more statistically powerful measure of genetic diversity.

## 3. Results

### 3.1. Arsenic

The phenotypic effects of As exposure on the *L. rubellus* used for the DNA methylation analysis in this study have previously been reported by Anderson et al. [18]. Mortality was not affected by As at any concentration. Significant effects of As on earthworm growth were found after 280 days exposure in the 36 mg/kg and 125 mg/kg treatments [18]. Of the total of 77 loci found using msAFLP analysis, 55 (71%) were methylation susceptible loci and 22 (29%) were not methylation susceptible as recorded from the protocol and analysis used. The respective proportions of the four types of msAFLP sites found can be seen in Figure 1. Pairwise genetic difference (фST) analysis showed no significant effects between the treatment for either the NML or MSL loci (фST = 0.0426; *p* > 0.05 and фST = −0.0216; *p* > 0.05, respectively) (Figure 1).

### 3.2. Cadmium

Cd significantly reduced juvenile survival at 500 mg/kg meaning that earthworms were not available for later growth measurements and for AFLP analysis for this concentration. *L. rubellus* growth was reduced by exposure to Cd at all tested concentrations. Divergent growth from the controls was evident for 148 mg/kg after 4 weeks exposure and for the 13 and 43.9 mg/kg treatments from 8 weeks onwards (See Figure 2). By the end of the 140-day exposure, the mean weight of earthworms was significantly lower than in controls for all treatments (ANOVA: 13 mg/kg, *p* < 0.05; 43.9 mg/kg, *p* < 0.01; 148 mg/kg, *p* < 0.001). Further, assessment of the clitellum formation indicated that maturation rates were also significantly reduced (*p* < 0.01) at all exposure levels. After 140 days exposure time, higher mortality had occurred in the 148 mg/kg treatment. Therefore, the exposure was truncated at this stage to ensure sufficient samples were available for the msAFLP analysis for this and all other concentrations.

From the msAFLP analysis, a total of 192 AFLP loci were detected in *L. rubellus* exposed at the different tested Cd concentrations. Of these loci, 127 (66%) were methylation susceptible and 65 (34%) were not methylation susceptible. Neither the AFLP nor MSL site comparisons indicated a significant difference between the exposed groups (фST = 0.03751; *p* > 0.05 and фST = 0.07266; *p* > 0.05, respectively) (Figure 3). However, a significant difference (χ^2^ = 9.2448, *p* < 0.01) in the proportions of type 4 sites (where no bands were detected either due to full methylation or a polymorphism) between the control and 13 mg/kg exposed earthworms was found.

### 3.3. Fluoranthene

There were no significant effects of fluoranthene exposure on survival at any of the tested concentrations. After 112 days exposure, no differences in average weights were observed. By the end of the exposure, the highest concentration (800 mg/kg) resulted in a significant negative effect on earthworm weights compared to the controls (ANOVA f-ratio = 23.7, *p* < 0.001). The negative effects at the highest concentration were associated with the change to freshly spiked soil after 112 days, indicating that over time fluoranthene had degraded in the initial set of spiked soils. The older (approximately mid-growth stage) juveniles showed a more pronounced response to the highest fluoranthene concentration than the freshly hatched juveniles. However, after this initial effect following the change of soils, the 800 mg/kg group slowly recovered, such that by 240 days, individuals at this treatment had reached similar average weights to controls. The 20.8 mg/kg exposure showed similar growth weights to the control group up until the 212 days and thereafter remained lower than in the other treatments, although this difference was not significant at the end of the exposure period (ANOVA f-ratio = 4.06, *p* > 0.05) (Figure 4). No other statistical difference in growth rate were evident. Further, there were also no significant impacts on maturation rates for any concentrations tested.

A total of 150 AFLP sites were found in the fluoranthene-exposed earthworms. Of these sites, 114 (76%) were methylation-sensitive and 36 (24%) were non-methylation sensitive. AFLP profiles showed no significant differences between treatments (фST = 0.0982; *p* > 0.05) (Figure 5). However, a significant effect of fluoranthene on msAFLP exposure was found (фST = 0.0351; *p* < 0.05). When comparing the proportions of the different types of AFLP sites, multiple significant differences were seen (Figure 6). Most notably, fluoranthene appeared to have an impact on the amount of Type 2, 3 and 4 sites. Effects on Type 2 and Type 3 sites showed contrasting trends, with Type 2 loci decreasing in frequency with concentration, while Type 3 loci frequency increased with higher fluoranthene concentration (Figure 6).

## 4. Discussion

Epigenetic factors are generally viewed as heritable changes in genomic function not accounted for by changes in either the coding or upstream promoter regions of a DNA sequence (2). There are a number of known mechanisms of epigenetic control, including DNA methylation, histone modifications and processes governed by the expression of small interfering RNAs (siRNA) and micro RNAs (miRNA) [9]. Recent findings indicate that these epigenetic mechanisms play a role in the regulation of gene expression that underpins the manner in which species, ecotypes and individuals respond to environmental perturbations [9]. As one of the main epigenetic mechanisms, studies indicate that the levels and patterns of DNA methylation can differ between and within different animal taxa [23]. Evidence indicates that annelids can have highly methylated genomes. For example, the lumbricid earthworm *Aporrectodea caliginosa trapezoides* has 13% of C residues as 5-methyl cytosine (m5C) (12). This is confirmed by our analysis here by msAFLP, which identifies a significant degree of DNA methylation in the genome of *L. rubellus*. Although the functional role of DNA methylation in annelids is not yet well established, observations of high levels of methylation point to a role as an important regulatory mechanism in the earthworm genome.

The results of the exposure and DNA methylation profiling illustrate some notable differences in the phenotypic and molecular responses of earthworms when exposed to different classes of toxic chemicals. For example, earthworm growth was affected at the highest As’ concentration (36 mg/kg per dry weight soil). However, the As exposure had no impact on DNA methylation status, as detected by msAFLP, and no AFLP genetic marker differences could be found between the treatment groups. In a study by Kille et al. [16], DNA methylation status was found to differ in populations of *L. rubellus* collected from sites around an As polluted mine with different As concentrations. The concentrations found in these mining-impacted soils were much higher than those used in the current experiment. For example, the on-site control used by Kille et al. [16], located away from the tailings area, contained 310 mg/kg As and the highest polluted soil contained 19,200 mg/kg As [16], as compared to the highest concentration used here for msAFLP analysis of 36 mg/kg. These higher levels in the mine soil may not necessarily be linked to greater As toxicity. It is likely that the speciation of the As in the mine spoil polluted soils may differ greatly from the As-III form used for the soil spiking. Studies of polluted soils have indicated a significant proportion is present in the As-V speciation state [24,25]. It is widely accepted that organic species in the As-V state are at least an order of magnitude less toxic than inorganic As-III species [26,27]. Hence, it seems likely that the mine site significant amount of AsV in mine sites’ soils may reduce the toxicity potential, meaning that the difference in toxicity expected from the As present in the mine site compared to laboratory soils is not as great as would be expected from total As concentrations.

Although the nature of the exposure may be less acute that expected from total As concentrations in the mine population, over the generational exposure times that have occurred for *L. rubellus* at this site, Kille et al. [16] found a relationship between soil As concentrations and population DNA methylation status. This association was not the case in the current study. The absence of DNA methylation change in the laboratory exposed, compared to the field collected *L. rubellus*, may identify that DNA methylation change is a mechanism linked more to long-term adaptive change than to short-term plasticity. Further, in the Kille et al. [16] study, the methylation differences were only observed when comparing lineage B individuals. In the current laboratory As-exposed earthworms, only 15 of 63 genotyped individuals were lineage B with the others being lineage A (as reported in Anderson et al., [18]. The lineage B individuals were spread across the treatments. Therefore, even if DNA methylation was altered in these individuals after life-time exposure, the number of lineage B representatives in each treatment is likely to be too small to clearly identify this pattern of response.

Exposure of juveniles to Cd affected earthworm growth at all exposure levels. Despite this effect, this metal had no impact on DNA methylation patterning, as indicated by the measurement using the msAFLP method. This exposure was truncated due to mortality, therefore different results may be reported if the exposure had been carried out for the full intended duration of 280 days. In another study, which used coelomocyte samples of the earthworm *Lumbricus terrestris*, an adult exposure to 10 mg/kg Cd resulted in significantly altered DNA methylation profiles in individuals exposed for 4 weeks. This difference was less apparent after 12 weeks in this study (as well as a replicate experiment reported in the same publication) [17]. Therefore, it is possible that DNA methylation in *L. rubellus* is indeed affected by Cd, but that this is a short-term response rather than a longer-term adaptive change. Further, any DNA methylation response may be restricted to adults, rather than juveniles, or perhaps is detectable when looking at coelomocytes, but not at whole tissue DNA.

Fluoranthene had no effect on the early growth of newly hatched juveniles and growth was only reduced following exposure to the highest concentration of 800 mg/kg in the later stages of the experiment, this time being 28 days after the worms had been placed in freshly spiked soil to account for fluoranthene degradation. Within the msAFLP analysis, the treatment groups separated significantly by methylation but not AFLP polymorphism profiles. A concentration-dependent effect on DNA methylation by fluoranthene was detected. The earthworms exposed to the highest soil concentration of 800 mg/kg, were most clearly separated from controls, followed by the earthworms exposed to 70 mg/kg, while those exposed to 20 mg/kg showed DNA methylation profiles that largely overlapped with those of the control group.

To our knowledge, an effect of fluoranthene on global DNA methylation status in a living organism has not been reported in any previous study. However, fluoranthene has been found to impact gene expression in adult *L. rubellus* exposed for 28 days [11]. Exposure to fluoranthene has also been associated with promoter methylation and different expression levels of cytochrome P4501A in chick embryos [28]. As DNA methylation has an impact on gene expression [29], it is possible that the changes in the DNA methylation profile seen may represent distinct molecular markers of fluoranthene exposure. It would need to be elucidated if fluoranthene has an effect on gene expression which in turn results in DNA methylation or if, alternatively, DNA methylation results in an alteration of gene expression (or a mixture of both). There is also evidence that fluoranthene can impact on DNA through limited adduct formation [30]. This DNA change may in turn alter the chromatin structure and, as a result, the DNA methylation status. Previous research on benzoapyrene has established that this PAH preferentially promotes DNA adduct formation in genome regions with high levels of cytosine methylation. It has been shown that prenatal exposure may have an impact on global DNA methylation [31]. Therefore, it is possible that fluoranthene elicits a similar effect in earthworms resulting in the changes in DNA methylation detected here.

## 5. Conclusions

In the present study, measurement of DNA methylation status after following exposure for the whole juvenile period identified some clear differences in earthworm methylome response to the different stressors. The study confirms the relatively high methylation status of the earthworm genome, providing further evidence of the potential role that DNA methylation could play in regulating phenotypic plasticity in this important group of invertebrates. Among chemicals, the polycyclic hydrocarbon fluoranthene was shown for the first time to have an effect on DNA methylation levels. Meanwhile, As and Cd did not show any effect on DNA methylation despite having been observed to elicit one in previous studies. To establish the role of DNA methylation in plasticity in earthworms, further studies are warranted. These studies may need to go beyond coarse methylation detection in whole body tissue samples to include whole genome and loci specific assessments in key tissues, such as the chlorogog and coelomocytes, known to be locations of pollutant accumulation and effect in earthworms.

This work was supported by the National Environmental Research Council through the GW4+ Doctoral Training Partnership NE/S007504/1 and NERC ERCITE grant award NE/S000224/2.

## Figures and Tables

**Figure 1 genes-13-00770-f001:**
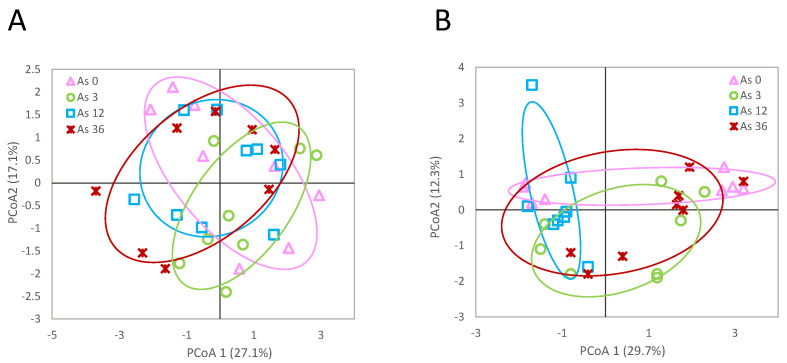
Principal Coordinate Analysis of the detected methylation (**A**) and genetic polymorphism (**B**) site patterns at the sequence 5′-CCGG-3′ for *L. rubellus* exposed for 280 days to a control and 3, 12 and 36 mg/kg per dry weight soil As throughout development from early juvenile stage.

**Figure 2 genes-13-00770-f002:**
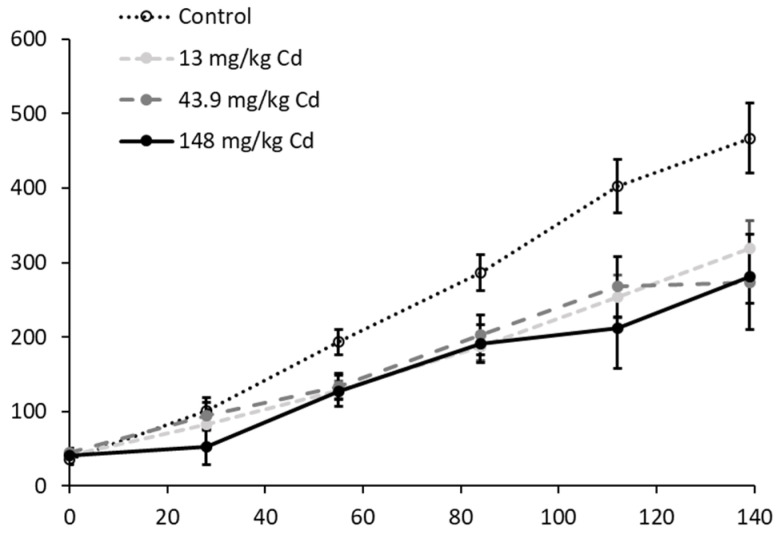
Mean ± SD weights (mg) of *L. rubellus* exposed to a control and three concentrations of Cd from a young juvenile stage throughout development for 140 days.

**Figure 3 genes-13-00770-f003:**
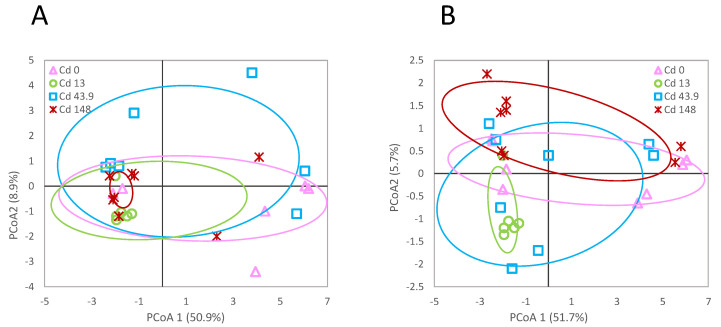
Principal Coordinate Analysis of the detected methylation (**A**) and genetic polymorphism (**B**) site patterns at the sequence 5′-CCGG-3′ for *L. rubellus* exposed for 140 days to a control and 13, 43.9 and 148 mg/kg per dry weight soil Cd throughout development from early juvenile stage.

**Figure 4 genes-13-00770-f004:**
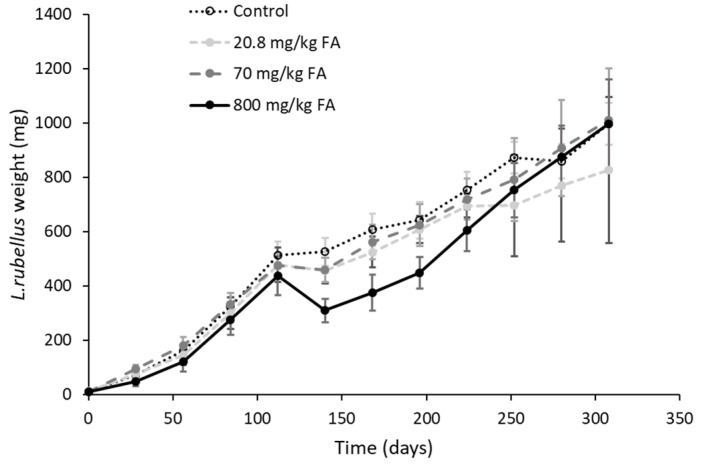
Mean ± SD weights (mg) of *L. rubellus* exposed to a control and three concentrations of fluoranthene from a young juvenile stage throughout development for 308 days.

**Figure 5 genes-13-00770-f005:**
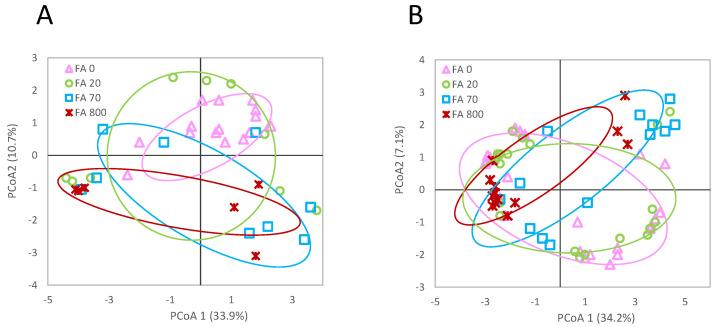
Principal Coordinate Analysis of the detected methylation (**A**) and genetic polymorphism (**B**) site patterns at the sequence 5′-CCGG-3′ for *L. rubellus* exposed for 308 days to a control and 20.8, 70, and 800 mg/kg per dry weight soil fluoranthene throughout development from early juvenile stage.

**Figure 6 genes-13-00770-f006:**
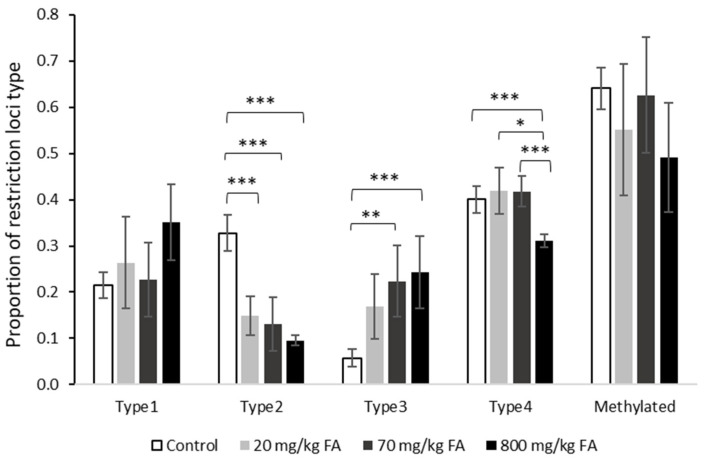
DNA methylation in methylation-sensitive loci detected from *L. rubellus* exposed for 308 days to a control 20.8, 70, and 800 mg/kg per dry weight soil fluoranthene; Types 1 to 4 are, respectively: Type 1. no methylation; Type 2. methylation of internal C; Type 3. methylation of external C or hemi-methylation; and Type 4. hypermethylation or mutation in restriction site (of the 5′-CCGG-3′ sequence). Global methylation level estimated following Nicotra et al., as proportion of (Type 2 + Type 3 loci/Type 2+Type 3 + Type 1 (scorable loci). Asterisks indicate significant between scorable loci types at * *p* < 0.05, ** *p* < 0.01, *** *p* < 0.001.

## Data Availability

Data available through Environmental Information Data Centre.

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
