# Peer review of "Impacts of Life-Time Exposure of Arsenic, Cadmium and Fluoranthene on the Earthworms’ L. rubellus Global DNA Methylation as Detected by msAFLP"

_genes, 2022, doi:10.3390/genes13050770_

Round 1

Reviewer 1 Report

The manuscript “Impacts of life-time exposure of arsenic, cadmium and fluoran-2 thene on the earthworms’ L. rubellus global DNA methylation 3 as detected by msAFLP” needs to pass major revision.

Abstract

Lines 17-18. Please rewrite this sentence

Lines 21-23. This is a sentence for a discussion…

Lines 26-28. Please rewrite they are hard to follow.

Introduction

Lines 65-66. Maybe “they exhibit” and not “the exhibits”

Mat and met.

Since all variables except msAFLP are acquired repeatedly from the same individuals please redo the statistical analysis with an statistical model that have into account lack of independence among measurements (repeated measures ANOVA, mixed models,…)

Results

Lines 217-222. If already published please do not include it in this manuscript. Otherwise, please give a table, figure or supplementary material to see data.

Line 226. ST should be defined in methods…

Line 237. F values (never seen before as “f”) has not any meaning without their degrees of freedom.

Figure 3. Please consider edit these figures to include all inertia ellipsis.

Discussion

Please consider to use the “Gene” bibliographic way of citation and not use numbers.

Author Response

 Abstract

Lines 17-18. Please rewrite this sentence. This sentence has been edited as suggested by the referee.

Lines 21-23. This is a sentence for a discussion… This sentence has been edited to ensure that it is now correctly placed in the result part of the abstract.

Lines 26-28. Please rewrite they are hard to follow. This sentence has been edited as suggested by the referee.

Introduction

Lines 65-66. Maybe “they exhibit” and not “the exhibits”. Corrected to “they exhibit” as suggested by the referee.

Mat and met.

Since all variables except msAFLP are acquired repeatedly from the same individuals please redo the statistical analysis with an statistical model that have into account lack of independence among measurements (repeated measures ANOVA, mixed models). The analysis of the survival and weight data was conducted only at the timepoint at the end of the experiment, this also being the same time that the earthworms were sampled for the msAFLP analysis. Hence, there would be no need for the use of statistical methods suitable for repeat measures data as only a single timepoint was included in the analysis. This has been clarified in the revised manuscript.

Results

Lines 217-222. If already published please do not include it in this manuscript. Otherwise, please give a table, figure or supplementary material to see data. We have edits this text to reduce the information only to a summary of major findings consistent with what would be done for any paper from which previous data was cited.

Line 226. ST should be defined in methods… ST clarified as “Pairwise genetic difference (ɸST)” as suggested by the referee

Line 237. F values (never seen before as “f”) has not any meaning without their degrees of freedom. This is a  good point, we have removed the details of f to provide prominence to the significance values.

Figure 3. Please consider edit these figures to include all inertia ellipsis. Figure have beeb redrawn as requested by the referee.

Discussion

Please consider to use the “Gene” bibliographic way of citation and not use numbers. No numbered citation has been included as suggested by the referee.

Reviewer 2 Report

The manuscript presents data of an extensive and long term study of effects of different chemicals on survival, growth, development and DNA methylation patterns in earthworms.

There are many spelling errors in the manuscript. The authors should read through and correct those. Some of them I listed here, but not all.

Line 19: The authors state here the concentration of 500mg/kg Cd, whereas in the material and methods they say that the used Cd concentrations were 0-148 mg/kg

Line 35: “for“ should be deleted

Lines 76-78: Please rephrase the sentence

Line 81: exposed instead of exposure

Line 91: write “and” instead of “of”

Some details are missing in the material and method section:

How many earthworms were exposed per treatment condition?

Did the authors extract DNA from each individual earthworm or pool of earthworms and how many earthworms were used per treatment for meAFLP?

Was DNA extracted using the whole earthworm or just a piece of tissue and if so which piece of the earthworm was used?

Line 227: Do you mean NML instead of AFLP loci?

Lines 309-312: Incomplete sentence, please rephrase

Line 316: The citation reference is missing

Line 319: write “not” instead of “no”

Line 370: write “term” instead of “tern”

Lines 413-416: Incomplete sentence, please rephrase

Author Response

The manuscript presents data of an extensive and long term study of effects of different chemicals on survival, growth, development and DNA methylation patterns in earthworms.

There are many spelling errors in the manuscript. The authors should read through and correct those. Some of them I listed here, but not all. The manuscript has been thoroughly proof read and check for flow and style to address the point raised by the referee.

Line 19: The authors state here the concentration of 500 mg/kg Cd, whereas in the material and methods they say that the used Cd concentrations were 0-148 mg/kg. Earthworms were exposed to 500 mg/kg, however, low survival at this treatment meant that individuals were not available for the later AFLP analyses. This has been clarified in the revised text. 

Line 35: “for“ should be deleted. “For” removed as suggested by the referee.

Lines 76-78: Please rephrase the sentence

Line 81: exposed instead of exposure

Line 91: write “and” instead of “of”. Edited as suggested by the referee.

Some details are missing in the material and method section.

How many earthworms were exposed per treatment condition? Number of earthworms (25) added as suggested by the referee.

Did the authors extract DNA from each individual earthworm or pool of earthworms and how many earthworms were used per treatment for meAFLP? Confirmation that DNA was extracted from the tissues of individual earthworms added to the text as suggested by the referee.

Was DNA extracted using the whole earthworm or just a piece of tissue and if so which piece of the earthworm was used? Confirmation that the analysis was conducted on powdered tissue obtained from whole earthworms added to the text as suggested by the referee.

Line 227: Do you mean NML instead of AFLP loci? Text corrected to NML as suggested by the referee.

Lines 309-312: Incomplete sentence, please rephrase. Sentence edited and ultimately shortened to give better clarity as suggested by the referee.

Line 316: The citation reference is missing. Citation added as suggested by the referee.

Line 319: write “not” instead of “no”. “No” change to “not” as suggested by the referee.

Line 370: write “term” instead of “tern”. “tern” changed to “term” as suggested by the referee.

Lines 413-416: Incomplete sentence, please rephrase. Sentence edited as suggested by the referee.